# The Multiple Origins of Roe Deer Populations in Western Iberia and Their Relevance for Conservation

**DOI:** 10.3390/ani10122419

**Published:** 2020-12-17

**Authors:** Tânia Barros, Eduardo Ferreira, Rita Gomes Rocha, Gonçalo Brotas, Juan Carranza, Carlos Fonseca, Rita Tinoco Torres

**Affiliations:** 1Departamento de Biologia & CESAM (Centro de Estudos do Ambiente e do Mar), Campus Universitário Santiago, Universidade de Aveiro, 3810-193 Aveiro, Portugal; taniabarros@ua.pt (T.B.); cfonseca@ua.pt (C.F.); rita.torres@ua.pt (R.T.T.); 2CIBIO-InBIO (Centro de Investigação em Biodiversidade e Recursos Genéticos), Campus de Vairão, Universidade do Porto, 4485-661 Vairão, Portugal; rgrocha@cibio.up.pt; 3ACHLI (Associação de Conservação do Habitat do Lobo Ibérico), Rua 25 de Abril 37, 4740-002 Esposende, Portugal; goncalobrotas@loboiberico.org; 4Wildlife Research Unit (UIRCP), Campus de Rabanales, Universidad de Córdoba, Colonia San José, 94560 Córdoba, Spain; jcarranza@uco.es; 5ForestWISE—Collaborative Laboratory for Integrated Forest & Fire Management, Quinta de Prados, 5001-801 Vila Real, Portugal

**Keywords:** admixture, allochthonous, *Capreolus capreolus*, genetic diversity hotspot, iberian peninsula, relict populations, pleistocene refugia

## Abstract

**Simple Summary:**

The Iberian Peninsula is considered a reservoir of genetic diversity and the source for the recolonization of continental Europe by several mammal species, after the last glaciation period. Here, we intended to characterize the genetic patterns and origins of one of those species, the roe deer, through the analysis of different genetic markers, as there is a major knowledge gap about the species’ populations at the western edge of roe deer distribution in the Iberian Peninsula. We found that western Iberia is a diversity hotspot for roe deer, with shared gene pools with other European and Iberian regions, but also with unique genetic elements, particularly the case of the relict population of Peneda Gerês National Park. Due to the distinct genetic diversity that was observed in western Iberian populations, we highlight the importance of these populations as sources of resilience against global changes. Our results provide useful information for the management and conservation of this species in the Iberian Peninsula. We advise transboundary management between Portugal and Spain as a rule, as well as careful evaluation of reintroduction actions, that should take in account the genetic data, in order to maintain the genetic heritage of roe deer in Europe.

**Abstract:**

The roe deer (*Capreolus capreolus*) is native and widespread in Europe and its phylogeography has been clarified in the last decades. Southern peninsulas are considered as reservoirs of genetic diversity and the source for the recolonization of Europe after the last glacial maximum. Even though roe deer populations have been genetically characterized, there is a major knowledge gap about the populations at the western edge of its distribution. To fill this caveat, and based on mitochondrial and nuclear DNA data, we aim to: (i) characterize the genetic diversity and structure of roe deer in western Iberia; (ii) clarify the origins and phylogeographical affinities of these populations, namely the relict population from Peneda Gerês National Park (PNPG, Portugal) and the likely allochthonous populations from central and south (CS) Portugal; (iii) discuss the implications of our findings for the management and conservation of the roe deer. Three major genetic clusters were inferred based on nuclear genotypes and were structured in a similar way as the three major mtDNA clades present in Iberia. Patterns inferred with nuclear markers confirmed PNPG as a relict population. Roe deer from CS Portugal share haplotypes with Central Europe rather than with other western Iberian populations, confirming its mainly allochthonous origin. Our results highlight western Iberia as a diversity hotspot for roe deer. We highlight the role of intraspecific genetic diversity as a source of resilience against ongoing global changes; the need for transboundary management and the importance of genetic data to inform management and conservation. When considered, repopulation or translocation measures should follow the IUCN Law of Reintroductions and meticulously conducted in order to preserve the genetic heritage of the species.

## 1. Introduction

The Mediterranean Basin is one of the very few World’s biodiversity hotspots located outside the tropics [1]. While the convergence of two biogeographic realms [2] contributes to this region’s singularity, the interaction among Earth’s climatic cycles and Europe’s geographic features plays a crucial role. At higher latitudes, Europe is a continuous landmass, but its southern ranges are composed of three—Iberian, Apennine and Balkan—Peninsulas, isolated, at some extent by mountain ranges along an E-W axis. It is nowadays well accepted that these peninsulas played a role as refugia and diversification centers, both at the inter and intraspecific level [3], namely for large mammal species, such as carnivores and ungulates [4].

The roe deer (*Capreolus capreolus*) is one of the most abundant and widespread wild ungulates in Europe. This species is native to the European continent and its distribution ranges from Scandinavian to the Balkans and Middle East, and to the Iberian Peninsula, at its southwestern distribution limit [5]. In the last 20 years, the study of the genetic patterns of the roe deer in Europe has been a subject of intense debate [6,7,8,9,10,11,12]. The species genetic structure, and the partition of genetic diversity among south peninsulas and central/northern Europe, supports the Pleistocene refugia hypothesis during the Last Glacial Maximum (LGM) [13,14]. Southern European peninsulas have been considered as high-diversity regions and likely sources for European recolonization [6,7], although a major contribution of one or more Eastern refugia has been suggested [15]. Post-LGM recolonization of Central and Northern Europe, from peninsular populations, has been postulated in other wild European ungulates, such as the red deer *Cervus elaphus* [16] and the wild boar *Sus scrofa* [17,18].

Southern European peninsulas are often highlighted as harbors for relict populations with high genetic diversity e.g., [7,12], and the Iberian Peninsula is no exception. Potential relict populations [7], and even exclusive genetic clades [8], have been identified for roe deer residing in North-western Portugal and Northern Spain, respectively. In fact, if the southern peninsulas acted as refugia during LGM, the presence of distinct gene pools and genetic structure is to be expected, due to the likely persistence of the original populations occupying those territories prior to the last glaciations, and populations entering Iberia, from the north, during the LGM [18]. Such a genetic structure and differentiation of Iberian roe deer populations was confirmed for extant populations in Spain [8,15].

Roe deer populations in Spanish territory were found to be genetically structured [15], and the observed phylogeographic patterns suggest the pre-existence of two principal contrasting groups that potentially arose due to the effects of climatic change in the Late Pleistocene, which covered the northern and central Iberian mountainous chains with ice. This scenario was also described for other mammal species such as shrews [19], rabbits [20], red foxes [21], and red deer [22]. For red deer, Carranza et al. [22] described two main lineages in Spain that predated the LGM, one in west and the other in central-south. In the case of the roe deer, the Iberian populations remained as a geographical isolate unit, with the Pyrenees acting as a potential geographical barrier, because neither the northern nor the southern haplotypes were found elsewhere in Europe [23]. This pattern was later confirmed by Royo et al. [8], who suggested the existence of two distinct refugia populations in Iberia: a north-western group, corresponding to *Capreolus capreolus decorus*, with a potential relict origin and with a strong divergence from the Iberian central-southern populations; and a southern group in Andalusia, corresponding to *C. c. garganta*, comprising only haplotypes belonging to the western European clade. Genetic differences were not enough for assigning a subspecies but were strong enough to validate specific management policies [8].

Much less is known about the populations in the western (Portuguese) side of Iberia. Randi et al. [7] highlighted the northwest region of Portugal (Peneda Gerês National Park, PNPG) as a potential relict population. The same authors recommended that this population, later assigned to *C. c. decorus* by Royo et al. [8], should not be artificially admixed with other roe deer populations, belonging to the sub-species *C. c. capreolus*. Nonetheless, no other study analyzed roe deer population at a wider national scale and therefore there is still a large knowledge gap on roe deer genetic diversity and phylogeographic relations of the populations at the westernmost edge of the Iberian Peninsula [7]. Given the role of Iberia as a faunal refugia, the high endemic genetic diversity of Iberian roe deer see [7,8], and the high genetic differentiation on other widespread ungulate species (e.g., red deer [22] and wild boar, [24]), it is of extreme relevance to disclose the genetic diversity and phylogeographic affinities of western Iberian roe deer populations, in order to inform present and future management and conservation practices.

Taking this into account, several questions remain unanswered about the origins of the western Iberia roe deer populations. Is the population from the north-western Portugal (PNPG) indeed a distinct relict population, as suggested by Randi et al. (2004)? Does this population distribution range extend to other areas in north-western Iberia? Or are the populations in northern Portugal the result of natural colonization from populations in north-western Spain [25]? On the other hand, what are the origins of the populations south of Douro river? It is often assumed, among academics, conservationists and managers, that extant roe deer native populations in Portugal are only present north of the Douro river. Populations on the south of this river are likely to be, at a great extent, the result of reintroductions of animals translocated from France and Spain during the 90′s [26]. Are the populations south of Douro river allochthonous, do they share an origin with Andalusian populations or are these populations distinct from any other Iberian population? This lack of knowledge raises concerns on how past conservation and management actions might have influenced the genetic makeup of roe deer populations in western Iberia, and how they may affect future management and conservation actions. To fill this caveat, we will use data on mitochondrial and nuclear DNA markers, in order to: (i) characterize the genetic diversity and structure of roe deer populations in western Iberia; (ii) clarify the origins and phylogeographical affinities of the PNPG population, and the likely allochthonous populations in previously unsampled areas in central and south Portugal (CS Portugal); and (iii) discuss the implications of our findings for the management and conservation of the Iberian roe deer.

## 2. Materials & Methods

### 2.1. Sample Collection

Roe deer samples were collected from different regions in the northern, central and southern Portugal, and from northern and north-western Spain (Appendix A). Spanish samples were retrieved from: wild populations from Galicia, Asturias and from a fenced population with individuals from NW Iberia (Valsemana, often used as source population for roe deer reintroductions in Iberia). Samples from north, central and south Portugal were retrieved from the Portuguese national collection of wildlife species tissue samples (Banco de Tecidos de Vertebrados Selvagens, ICNF—Instituto da Conservação da Natureza e das Florestas) and also collected from hunted and/or collected dead animals. Muscle samples were stored in 96% Ethanol at −20 °C and blood samples in FTA^®^ cards stored at room temperature for posterior usage. A total of 116 samples were collected and submitted to DNA isolation.

### 2.2. Laboratory Procedures, Sequencing and Genotyping

DNA was isolated from blood and tissue samples using the salt extraction method [27]. DNA extraction from trophies (antlers) was performed using JETFLEX^®^ kit from Genomed^®^, following the manufacturer’s recommendations. Samples were also genotyped for 9 microsatellites with autosomal origin: CSSM66, INRA006, BM6506, BM1818, BM757, NVHRT48, NVHRT21, RT1, NVHRT16 [8]. Primers were marked with fluorescent dyes (6-FAM, HEX and NED), in order to allow simultaneous fragment analysis of several markers. Microsatellites were amplified using QIAGEN^®^ Multiplex PCR Kit™ with a total volume of 20ul of reaction mix and 2ul of extracted DNA, following manufacturers conditions, in three multiplex sets (M1: NVHRT48, NVHRT16, CSSM66; M2: INRA006, RT1, BM6505; M3: NVHRT21, BM1818, BM757). Success of PCR multiplex amplification was verified in an agarose gel. Successfully amplified products were enzymatically purified and electrophoresis of PCR products (fragment analysis) was conducted in an automatic sequencer ABIPRISM^®^ 3730-XL DNA Analyzer (Applied Biosystems™). Electrophoretograms were analyzed using Peak Scanner™ v1.0 (Applied Biosystems™) and allele calling was performed manually. Because DNA from antlers is often present in low quantity and quality, it was not possible to retrieve enough data from all samples. Thus, we only considered genotypes for which information for at least eight of the nine microsatellite markers was available.

A fragment of d-loop region (mitochondrial DNA) was amplified using the primers L-Pro e H-16493 [6,28]. PCR reactions were performed on a final volume of 20 μL (1–2 μL of DNA isolate), using INVITROGEN^®^
*Taq* DNA Polymerase (following the manufacturer’s conditions) and an annealing temperature of 60 °C. PCR products were visualized in agarose gels, purified and sequences in an automatic sequencer ABIPRISM^®^ 3730-XL DNA Analyzer da Applied Biosystems™. Mitochondrial DNA sequences were aligned using CLUSTALW algorithm on MEGA 5 [29], with subsequent manual alignment.

## 3. Data Analysis

### 3.1. Genetic Structure and Diversity Assessment Using Nuclear Markers

The inference of genetic structure and individual probability of assignment of each individual to each genetic cluster was performed, without any a priori information, on STRUCTURE v2.3.4 [30,31]. Results were analyzed on STRUCTURE HARVESTER [32] and best K was estimated using the Evanno method [33]. Population structure and admixture may cause substantial deviations in the numbers of homozygotes and heterozygotes. Thus, genotype data were tested, a posteriori, for each population, for significant evidence of bias caused by the presence of null alleles, allele dropout or stutter, with Micro-checker [34] Genetic distances between sampling regions (defined a priori: Alentejo, Beiras, Peneda Gerês National Park—PNPG, Trás-os-Montes, Galicia, Asturias and Valsemana, Figure 1 and Appendix A) and, among populations (defined *a posteriori,* based on genetic diversity and structure results: CS Portugal, PNPG, Trás-os-Montes, NW Iberia and Valsemana) were estimated following Slatkin [35] and Nei [36]. Samples were grouped into populations based on genetic cluster assignment probabilities and the estimated genetic distances among sampling regions. Deviations from Hardy Weinberg equilibrium and inbreeding coefficient (FIS) were estimated on ARLEQUIN v3.5 [37]. Calculation of general diversity indices (number of alleles, effective allele size and private alleles) and Principal Coordinate Analysis (PCoA) were performed on Genalex 6.5 [38]. In order to search for evidences of recent bottlenecks, we calculated the values of M ratio values [39]. We estimated the degree and direction of asymmetric gene flow among populations using the relative migration network method developed by Sundqvist et al., 2016, which is implemented in the function *divMigrate* of the diveRsity R package [40]. Significant asymmetric migration flow was estimated based on a bootstrap procedure with 50,000 replicates. Estimates of migration rates using this method are based on allele frequencies differences and relative migration coefficients were estimated using the number of migrants (Nm) parameter [41].

### 3.2. Phylogenetic Structure and Affinities with Iberian and European Populations

Mitochondrial D-loop sequences generated in this study were used to infer mitochondrial haplotypes present in NW Iberia. Number of haplotypes was inferred using DnaSP [42]. Mean genetic distances between individuals from the same population and between different populations were estimated following Tamura and Nei using MEGA [43]. Number of haplotypes, haplotype diversity, nucleotide diversity and number of mutations among haplotypes were assessed on DNAsp v5 [44]. In order to infer the phylogeographic relationships among haplotypes sampled in western Iberia, and their distribution across populations, we built a medium-joining network of the haplotypes generated in this study, using PopArt [45]. Additionally, a second network was built with the aim of contextualizing Iberian sequences in a European framework, using previously published sequences from other European regions (Appendix A). A total of 572 d-loop sequences were retrieved from GenBank. We divided the sequences according to the five populations defined a posteriori and we added the following groups: Italy (corresponding to the Apennine peninsula), Balkans (Bulgaria, Greece, Serbia, Romania and Slovenia) and Central/Northern Europe (France, Germany, Austria, Hungary, Poland, Slovakia, Denmark, Finland, Norway and United Kingdom).

## 4. Results

### 4.1. Genetic Structure and Diversity

Individual nuclear genotypes were retrieved from 108 samples and, based on these genotypes (Appendix A), a partition on three genetic clusters (K = 3) were inferred as the most likely pattern of population structure (Figure 1A,C). The three inferred clusters present a strong agreement with the geographic origin of samples and with previous knowledge on roe deer populations: one of the clusters (PGNP) was strongly associated (average individual proportion of assignment *q* = 89%) with individuals sampled in the putative relict population of Peneda Gerês National Park; a second cluster (CSP) was strongly associated with individuals from the putative allochthonous populations south of Douro river (*q* = 88% both in Alentejo and Beiras); the third cluster (NWI) was mostly associated with individuals from north-western Spain (*q =* 60% in Asturias, 66% in Galicia and 84% in the fenced population of Valsemana) and, at a lesser extent, northeastern Portugal (*q* = 50% in Trás-os-Montes). Mean genetic distances, estimated from microsatellite data (Appendix A) revealed that genetic distances among sampling locations are in general significant, except for the pairs Alentejo/Beiras and Astúrias/Galicia, where distances were systematically nonsignificant. Evidences of significantly asymmetric gene flow (Figure 1C) were found from the population of Peneda Gerês National Park to Beiras (CS Portugal) and to Trás-os-Montes (although in the latter, it was only marginally significant). Based on assignment probabilities and genetic distances, and the fact that Valsemana is an enclosed population, we defined five populations that were considered henceforth (Figure 1C): Central and South Portugal (CS Portugal), Peneda Gerês National Park (Peneda Gerês), Trás-os-Montes, Valsemana and North-western Iberia (NW Iberia). Pairwise genetic distances estimated for the five populations were always significant (Table 1). When testing each of the five poulations for significant bias introduced by the presence of null alleles, allele dropout or stutter (Appendix A), we found no evidence for systematic bias in genotyping. No evidence at all was detected for the two largest samples (PGNP and NW Iberia). Significant excess of homozygotes—which can be interpreted as evidence of null alleles—was found in only three markers (CSM66, RT1 and BM1818), but only in one or two populations. Valsemana is a fenced population with high levels of inbreeding and Trás-os-Montes and CS Portugal are populations with evidence for asymmetric incoming gene flow. Given the lack of evidence for systematic bias and the likely biological meaning of the homozygote excess, we decided to include all loci in the analyses.

Significantly lower heterozygosity relatively to the expectations under Hardy-Weinberg equilibrium conditions were found only for CS Portugal, Peneda Gerês and Valsemana (Table 2), and only in two to three out of nine markers. Genetic diversity, numbers of alleles, as well effective and private alleles were always higher in CS Portugal and lower in Peneda Gerês. Inbreeding coefficient was lower in NW Iberia and higher in the fenced population of Valsemana. Estimated M ratio values were generally high, and only the value for Peneda Gerês was slightly below the theoretical minimum value for equilibrium populations estimated by Garza and Williamson [38]. The pattern revealed by the ordination (PCoA, Figure 2) of individual genotypes is consistent with estimated genetic distances, with individuals from Peneda Gerês and CS Portugal being the most dissimilar among each other and an overall overlap among samples from the enclosed population of Valsemana and NW Iberia.

From the sequencing of mitochondrial DNA, we were able to retrieve a 480 bp length sequence for 111 samples. A total of 19 haplotypes were identified across all sampling regions (Table 3). Diversity indices estimated for mtDNA were higher in NW Iberia and CS Portugal and lower in Trás-os-Montes and Peneda Gerês (Figure 1B and Table 3) with more haplotypes, higher haplotype and nucleotide diversity and average number of mutations among haplotypes in the first two populations. With exception to East clade, all major clades (Central—including the Iberian subclade, West and Celtic Iberian) identified by Randi [7] and Royo [8] were found in the study area. The clades were geographically structured and showed strong overlap with the genetic clusters inferred from nuclear markers. The major difference was found for Trás-os-Montes and Galicia. While individual genotypes from these two sampling areas were assigned with higher posterior probability to the NWI cluster, together with Astúrias and Valsemana, the most frequent haplotypes—in particular in the Trás-os-Montes population—were mainly from the Central European clade, though from the Iberian subclade. Haplotypes H15 and H9/H14 (Celtic-Iberian clade) were found exclusively in Valsemana and NW Iberia (Table 3), while 6 haplotypes (Central European clade: H2, H6, H16, H17, H18; West clade: H19) were exclusive from CS Portugal. Haplotype H1, from the Iberian subclade, was found to be widely distributed across the study area and H8 (West clade) was the most frequent haplotype, mostly found in Peneda Gerês and, at a lesser extent, in CS Portugal.

### 4.2. Phylogeographic Relationships in Iberia and Europe

When considering only the haplotypes sampled in Western Iberia, it becomes clear that haplotypes from different clades are to some extent geographically structured (Figure 3 and Table 3), even if not completely segregated. Most of the haplotypes assigned to West clade were sampled in Peneda Gerês National Park (32 out of 43 individuals, 74%) or in NW Iberia (6 individuals, 14%). Haplotypes assigned to Celtic-Iberian clade were exclusively found in the northern part of the study area outside Peneda Gerês, in Valsemana (12 out of 21 individuals, 57%), NW Iberia (7 individuals, 33%) and Trás-os-Montes (2, 10%). However, Trás-os-Montes was the region were the haplotypes belonging to the Iberian subclade of the Central clade were more frequent (11 out of 23 individuals, 48%), followed by NW Iberia (7 individuals, 30%). The large majority of haplotypes from Central clade, not related with the Iberian subclade, were found in CS Portugal (18 out of 24 individuals, 75%).

The integrated phylogeographic analysis comprising the haplotypes generated here and the ones retrieved from GenBank, resulted in an alignment of 154 haplotypes from across Europe, with a sequence length of 388 bp. Haplotype designation from H1 to H19 in the European network (Figure 4) matches the designations in the network inferred from the newly generated sequences (Figure 3). Haplotype H16 collapsed with H10, after the trimming of sequences. General patterns are in agreement with patterns referred above, for our study area. From Central clade, haplotypes H1 and H7 appear as exclusively Iberian, and shared among different Iberian regions, while other haplotypes (H2, H6, H10/H16 and H13) are found outside Iberia, in other European regions. Celtic Iberian haplotypes that were also exclusively found in Iberia (H9, H12, H14, H15), with exception to H5, also sampled in Central and North Europe, are still unique haplotypes from NW Iberia, while haplotypes H17, H18 (Central clade) and H19 (West clade) correspond to singletons found in CS Portugal (Figure 4).

However, all haplotypes from Central clade, sampled in more than one individual in CS Portugal, were shared with individuals sampled in Central and North Europe. Several other Central clade haplotypes sampled in Iberia in this (H13) or other studies (H20, H24, H96) where found to be shared with other European regions. On the other hand, while haplotypes from West clade were also found elsewhere in Europe, all haplotypes assigned to this clade, that were sampled in Iberia—including H8 (which was reported for three individuals sampled in CS Portugal)—were exclusively sampled in this peninsula. No haplotypes from East clade (dominantly from Balkan origin) were sampled in Iberia, based on available information.

## 5. Conclusions

The Iberian roe deer population is highly diverse and genetically structured, with elements characteristic of three of the main European clades: West, Central, and Celtic-Iberian this study [7,8,9,15,46]. Here, we update and expand the knowledge to the westernmost populations of Iberian Peninsula, supporting multiple origins for this roe deer population. In fact, all major clades present in western Europe [7] are represented in the western Iberian Peninsula, suggesting a diversity hotspot within the Iberian hotspot: the Celtic-Iberian clade, an exclusively Iberian element; the West clade, a clade with origin in Iberia, as suggested by Randi et al. (2004); and the exclusively Iberian subclade within Central clade, which may have reached Iberia secondarily, as suggested by Royo et al. [8].

### 5.1. Western Iberia Populations—Their Structure and Origins

The structure and diversity patterns inferred from nuclear DNA highlight the distinctiveness of the roe deer populations in Peneda Gerês, thus corroborating the hypothesis advanced by Randi et al. [7] that this was in fact a relict population of the pre-LGM Iberian gene pool. The high dominance of West clade haplotypes—a clade with likely Iberian origin (Randi et al. 2004)—in Peneda Gerês, is only paralleled by its dominance in South Spain, and specifically Andalusia, as previously reported by Royo et al. [8]. Roe deer from Andalusia is also considered a distinct and possible relict population [8,23] that has received the status of subspecies (*C. capreolus garganta*, [47]), despite the lack of consensus on the subspecific status [48]. Lorenzini & Lovari [15] analysed sequences from 14 allegedly native roe deer European populations, including samples from PNPG, in order to prevent misleading phylogeographical conclusions. These authors also inferred a panmictic Pleistocene ancestral origin hypothesis of the native Portuguese northern populations, confined to that area due to climatic changes and human influence, that was supported by mitochondrial DNA data. However, biparental markers indicate a recent admixture between those Portuguese roe deer and northern and southern Spanish roe deer, most likely attributed to male roaming. Because the lack of sex-biased dispersal in roe deer e.g., [49,50], it is unlikely that close populations harbor only mutually exclusively mtDNA haplotypes or nuclear alleles. Therefore, we consider that the strong dominance of distinct haplotype groups and posterior probabilities of assignment are very strong evidences of the distinctiveness of this population. The dominance of West clade haplotypes leaves few doubts on the Iberian origin of this population, that should thus be a relict population.

Likewise, both the structure and diversity patterns inferred from nuclear and mitochondrial DNA identify the populations south of Douro river (CS Portugal) also as distinct from other western Iberian populations. Contrarily to populations in southern Spain, which are dominated by West clade haplotypes [7], CS Portugal is dominated by Central clade haplotypes. Dominance of Central clade haplotypes is particularly frequent in Iberian populations known to have suffered human-mediated translocations in the recent past, namely CS Portugal [26] and North-eastern Spain [8,51] and might be interpreted as an evidence of allochthonous origin. The presence of shared haplotypes among European regions could also be viewed as an evidence of migrations occurred during the last glacial maximum [52], but such shared haplotypes are often among the most frequent ones and occupy central positions in phylogeographic networks. This is not the case of the shared haplotypes found in CS Portugal. We therefore consider that available information supports the hypothesis that CS Portugal is mainly derived from recent human-mediated translocations of roe deer from France, and thus of allochthonous origin. Nevertheless, haplotypes belonging to the Iberian subclade and even to West clade (including an exclusive haplotype: H19) were sampled in this population. Thus, there are native Iberian elements in this population, suggesting that there might still be remnants of the original roe deer populations from Centre and South Portugal, that became admixed with the translocated individuals. While no individual from this population was assigned with high posterior probability to a genetic cluster different from CSP does not conflict with this hypothesis, for—as mentioned above—genotypes are reconstructed de novo at each new generation but mtDNA is non-recombinant and holds the evidences for a longer time. Nevertheless, we did find evidences of asymmetric gene flow from PGNP towards CS Portugal. Additionally, the higher numbers of alleles, haplotypes and genetic diversity, as well as mean distances among haplotypes in the CS Portugal population is also consistent with an admixed origin and thus provides further support to the view that this is a mainly allochthonous population admixed with remnants of the native populations.

Other populations show a mostly admixed origin, such as the case of Trás-os-Montes and NW Iberia (comprising Asturias and Galicia). The existence of shared haplotypes between regions from northern Portugal and north-western Spain corroborates the hypothesis of natural colonization of individuals from transboundary population [25]. Populations from Trás-os-Montes and North Western Iberia are characterized by higher frequencies of haplotypes from West, Celtic-Iberian and Central clades. However, and contrarily to CS Portugal, Central clade haplotypes in those populations belong either to the Iberian subclade (H1 and H7) or are widespread, abundant and nuclear haplotypes in Central clade (H24, H13 and H20). This is consistent with the existence of shared ancestral polymorphism, as it was already hypothesized for other distributional ranges of roe deer in Europe [9] and observed in the case of wild boar [52].

The Celtic-Iberian haplotypes were the most common in the fenced population of Valsemana and also in Asturias, in line with what was observed by Royo et al. [8], but curiously not in the north-western corner of Iberia. While the results are in line with the enclosed nature of the Valsemana population, they also confirm that this population was founded using the regional gene pool, given the high dominance of haplotypes from the Celtic-Iberian clade. In fact, the frequency of the different clades is very similar to that of the individuals sampled in Asturias. While a higher level of inbreeding and low level of heterozygosity is expected in an enclosed population, diversity levels—in particular in the case of nuclear markers—are among the highest in the studied populations.

Structure and diversity patterns inferred with nuclear markers were consistent with the patterns observed in mtDNA. Concordant structure patterns inferred from mtDNA and microsatellite data were previously reported by Biosa and collaborators [11], for central Italy. Both roe deer males and females are likely to disperse at around 1-year age but after their post-dispersal range, adult individuals display high site fidelity [53,54]. Sex-biased dispersal towards males promotes gene flow of nuclear but not mitochondrial alleles. However, low sex bias in dispersal rates has been reported for roe deer e.g., [48,49]. While the geographic distribution of the genetic clusters PGNP and CSP largely overlaps with the distribution of West and Central clades, respectively, in the study area, a much lower overlap exists between cluster NWI and Celtic-Iberian clade. Nevertheless, the variation on the average posterior probability of assignment matches the variation in the frequency of Celtic-Iberian clade (higher in Valsemana and lower in Trás-os-Montes). Three distinct processes might have a role on the observed mismatch. First, mitochondrial genes are non-recombinant markers so evidences of contributions of different gene pools are less likely to dilute in the population, than in the case of genotypes, that are reorganized every new generation. Additionally, founder events and genetic drift are more likely to influence the differentiation of microsatellite nuclear markers than of mitochondrial markers. Lastly, Trás-os-Montes (and, at a lesser extent, Galicia) are located at the convergence of distinct populations and are thus, more likely to receive distinct genetic contributions and present a more admixed origin. While gene flow appears to be mainly symmetric within the studied populations, asymmetric gene flow from the population of Peneda Gerês to Trás-os-Montes and to Beiras (that is part of CS Portugal population) was indeed confirmed. Nevertheless, more evidence (e.g., more intense sampling effort) might be needed to clarify whether these populations—particularly Trás-os-Montes—are mainly admixed or might correspond yet to a fourth genetic cluster matching the dominance of a distinct set of haplotypes.

Genetic differentiation is observed both at a continental and a regional scale, and non-intuitive barriers to gene flow occur. We believe this pattern is also patent in the mitochondrial signature of roe deer in the Iberian Peninsula, as the majority of the haplotypes from Iberian Peninsula are exclusive from this region. Similarly, unique haplotypes are found in other geographic regions, including Italy and Balkans, the last being related with glaciation periods [14,55,56]. Our results showed that roe deer populations in western Iberia harbour a great deal of diversity and a set of haplotypes not found elsewhere in Europe. As in the case of red deer [22] and wild boar [52], Iberian roe deer populations have a large component of unshared diversity but also shared elements with other European regions, particularly with Northern post-glacial re-colonizers in the case of red deer and with Central Europe in the case of wild boar. In the case of roe deer, original Iberian lineages are dominant in more peripheral regions (north-western and south Iberia) and shared components are more frequent in the regions closer to Central Europe (such as north-eastern Spain), or in regions that suffered recent human-mediated translocations (such as Central and South Portugal).

### 5.2. Management and Conservation Implications

Iberia is a genetic diversity hotspot for roe deer [7,8,15], being the western Iberia a diversity hotspot within the Iberian hotspot, as previously shown for other species [18,57]. North-western Iberian populations are dominated by haplotypes from three major (Central, West and Celtic-Iberian) clades. Estimated divergence times among the three clades largely precedes the last glacial maximum, ranging from 98-35kya or 91-45kya—between Central and West or Celtic-Iberian clades, respectively—to 221-111kya—between Celtic Iberian and West clade [8,23]. Such an old diversification suggests that roe deer might have arrived and diversified within Iberia more than once. The accumulation of genetic diversity (and often associated local adaptations) is a key element on species resilience to ongoing global (land-use and climate) changes [58]. This is especially relevant since the whole ecology of this species at the southern end of its distribution is poorly known. In the perspective of the ongoing climatic changes in the Mediterranean area, we highlight the fundamental importance of this population preserving a unique genetic diversity.

As pointed by Royo et al. [8], genetic structure patterns match, to some extent, the distribution of roe deer subspecies described at the beginning of the last century, based on morphological criteria. Whether or not *C.c garganta*, *C.c. canus* and *C.c. decorus* are valid subspecies is not debated here. However, the matching between genetic and morphological variation may be an important clue to ecologically relevant differences and adaptations to different climatic regions, terrains or land-uses. As rewilding initiatives are proliferating in Europe, genetic diversity patterns can be used as a relevant proxy for choosing suitable populations for translocations, within a conservation context. Captive-reproduced stocks of ungulates are common in some regions of Europe e.g., [59], but choosing a source population genetically different and/or from different ecological conditions, might result in the disruption of the gene pool and local adaptations of target populations [60,61]. According to the Law of Reintroductions from IUCN (1998), source populations must be genetically related with the native population and show similar ecological characteristics. For example, our results show that, despite the widely diverse Iberian roe deer gene pool, historical translocations in central and south Portugal [26] have introduced an allochthonous gene pool, distinct from any Iberian population. The extent to which these translocations have disrupted any remaining original populations and local adaptations is unknown. Another relevant insight provided here and in previous studies [18] is that populations and gene pools are not restricted to administrative limits, often crossing national borders. Gene flow and admixture among roe deer populations across the Portuguese-Spanish border seems to be the rule, rather than the exception. As in other medium to large size mammal species, management and conservation actions taken in one side of the border have an influence on the whole population, interpreted as a reproductive unit. Transboundary management and conservation should thus be the rule, rather than exception.

## Figures and Tables

**Figure 1 animals-10-02419-f001:**
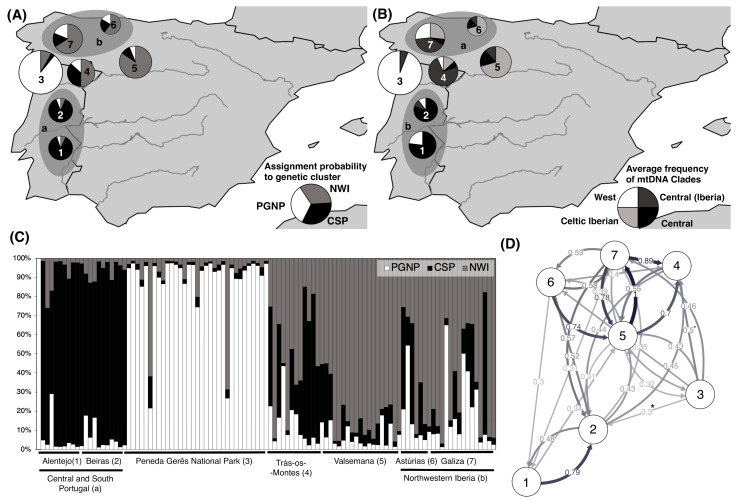
Roe Deer genetic diversity, structure and gene flow in Western Iberia. (**A**) Population average posterior probabilities of assignment to genetic clusters (PGNP, NWI, CSP), inferred from microsatellite genotype data. (**B**) Population frequency of mtDNA haplotypes belonging to Central (and Iberian subclade), West and Celtic-Iberian clades, after Randi et al. (2004) and Royo et al. (2007). Circle sizes are proportional to sample size. (**C**) Individual posterior probabilities of assignment to inferred genetic clusters (PGNP, NWI, CSP). (**D**) Relative migration flows among populations in number of migrants, Nm. Significantly asymmetric migration flow (i.e., gene flow) marked with an asterisk (*—*p* < 0.05) or a dot (▪—*p* < 0.1). Sampling areas: 1—Alentejo; 2—Beiras; 3—Parque Nacional Peneda Gerês; 4—Trás-os-Montes; 5—Valsemana; 6—Asturias; 7—Galica; a—Central and Southern Portugal (1 + 2); b—Nortwesten Iberia (7 + 6).

**Figure 2 animals-10-02419-f002:**
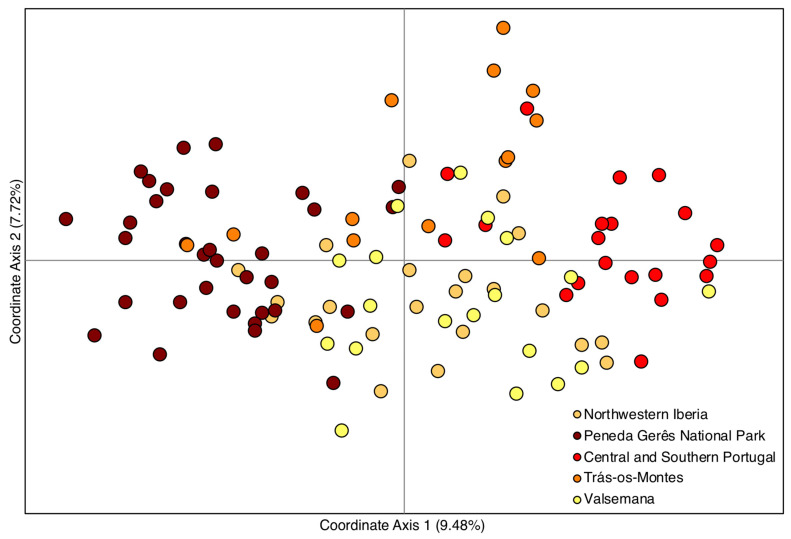
Relations among individual genotypes from the five sampled populations. Two-dimensional plot based on the first two variation axes of principal coordinate analysis. Percent of explained variation is reported for each axis. Individual genotypes are color-coded, according to source population.

**Figure 3 animals-10-02419-f003:**
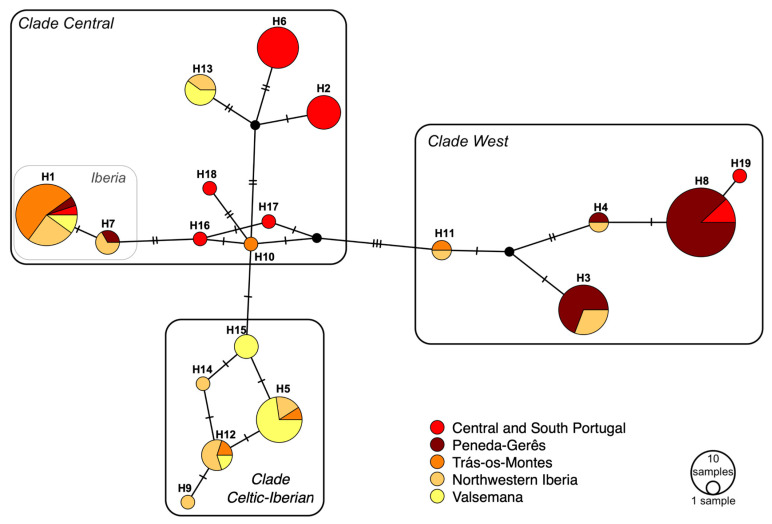
Median-joining network depicting the phylogenetic relations among haplotypes sampled in this study. Circle size is proportional to haplotype frequency. Circles are colored according to haplotype frequency in each area. Haplotypes clustering within Central (including Iberian subclade), West and Celtic-Iberian clades (following Randi et al., 2004; Royo et al., 2007) are delimited. Each dash among haplotypes represents a mutational step. Black nodes stand for hypothetical (not sampled) ancestral haplotypes.

**Figure 4 animals-10-02419-f004:**
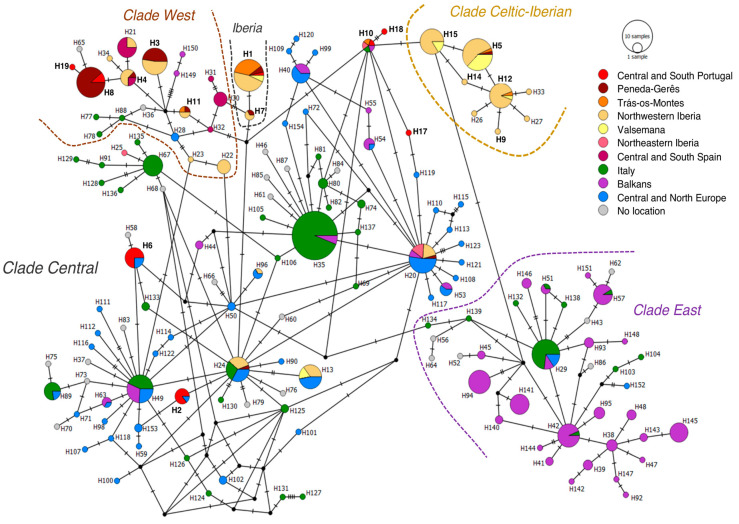
Median-joining network depicting the phylogenetic relations among haplotypes sampled in this study and haplotypes from all Europe retrieved from Genebank. Circle size is proportional to haplotype frequency. Circles are colored according to haplotype frequency in each area in western Iberia and Europe. Haplotypes clustering within Central (including Iberian subclade), West and Celtic-Iberian clades (following Randi et al., 2004; Royo et al., 2007) are delimited. Each dash among haplotypes represents a mutational step. Black nodes stand for hypothetical (not sampled) ancestral haplotypes.

**Table 1 animals-10-02419-t001:** Genetic distances among populations within genetic Iberia. Lower left matrix diagonal—Slatkin’s distance estimated from microsatellite genotype data. Upper right matrix diagonal—Mean number of substitutions per site (Tamura-Nei model) estimated from mitochondrial (D-loop) haplotypes. All distances were significant (*p* < 0.05).

	Central & South Portugal	Peneda Gerês	Trás-os-Montes	NW Spain	Valsemana
Central & South Portugal		0.012	0.011	0.012	0.011
Peneda Gerês	0.170		0.014	0.014	0.017
Trás-os-Montes	0.091	0.105		0.011	0.011
North-western Spain	0.084	0.069	0.051		0.011
Valsemana	0.121	0.117	0.065	0.033	

**Table 2 animals-10-02419-t002:** Diversity indices estimated from nuclear and mitochondrial markers, for roe deer populations in Western Iberia. CS Portugal: Central and South Portugal; NW Spain: North-western Spain. N: number of samples; H_OBS_ and H_EXP_—Observed and expected heterozygosity; within brackets, the number of markers with significant deviations from Hardy-Weinberg equilibrium conditions, after Bonferroni correction; N_a_, N_effect_ and N_priv_—Average number of alleles, effective allele size and private alleles; M—M ratio (Garza & Williamson, 2001); F_IS_—Inbreeding coefficient; H—Number of haplotypes; h—Haplotype diversity; Π—Nucleotide diversity; k—Average number of mutations among haplotypes.

Population	Nuclear DNA (Microsatellites)									Mitochondrial DNA (D-loop)						
	N	H_OBS_	H_EXP_	Alleles (Average Values)				M	F_IS_	Genetic Diversity	N	H	h	Polymorphic Sites	Π	k
				N_a_	N_effect_	N_priv_	Range									
CS Portugal	20	0.626	0.662 (2)	6.67	4.20	0.67	18.6	0.731	0.098	0.661	23	8	0.787	14	0.0081	3.91
Peneda Gerês	33	0.545	0.580 (3)	4.56	2.74	0.44	14.7	0.622	0.067	0.547	34	5	0.524	10	0.0049	2.35
Trás-os-Montes	14	0.647	0.728 (0)	5.70	3.60	0.22	15.9	0.709	0.053	0.573	15	5	0.476	10	0.0055	2.63
NW Spain	22	0.625	0.671 (0)	6.00	3.66	0.44	16.9	0.707	0.032	0.632	22	10	0.905	17	0.0116	5.55
Valsemana	19	0.560	0.653 (2)	6.00	3.60	0.56	18.0	0.722	0.154	0.647	17	5	0.743	10	0.0065	3.12

**Table 3 animals-10-02419-t003:** Roe deer haplotype (mtDNA) frequency and distribution by sampling region (and population) in Western Iberia. Haplotypes are organized according to their assignment to mtDNA clades inferred by Randi et al. [7] and Royo et al. [8]: Central. West. Celtic-Iberian. Haplotypes clustering within Iberian subclade from Central clade (Randi et al. [7]) are also identified.

Haplotype	Central & South Portugal		Peneda Gerês	Trás-os-Montes	North-western Iberia		Valsemana
	Alentejo	Beiras			Asturias	Galicia	
**Clade Central**							
*H2*	1	5					
*H6*	7	2					
*H10*				1			
*H13*					1	1	3
*H16*		1					
*H17*	1						
*H18*	1						
**Clade Central (Iberian subclade)**							
*H1*		1	1	11	1	4	2
*H7*			1			2	
**Clade West**							
*H3*			9		1	3	
*H4*			1		1		
*H8*	2	1	22				
*H11*				1		1	
*H19*	1						
**Clade Celtic-Iberian**							
*H5*				1	1	1	8
*H9*						1	
*H12*				1	2	1	1
*H14*						1	
*H15*							3
***Total***	***13***	***10***	***34***	***15***	***7***	***15***	***17***

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
