# Peer review of "The Multiple Origins of Roe Deer Populations in Western Iberia and Their Relevance for Conservation"

_animals, 2020, doi:10.3390/ani10122419_

Round 1
Reviewer 1 Report
Thank you for these new elements on the genetics of Portuguese deer populations. You will find some remarks in the text.

Author Response
Reviewer 1
We have incorporated all the suggestions made by the reviewer. Because we also included some of the suggestions made by the second reviewer, the line numbers in the current version of the paper had changed. Nevertheless, in order to allow easy tracking by reviewer 1, as reference for comment location, we used the line number of the original pdf version with reviewer 1 comments and suggestions. Within brackets, we include the location of the modification in the new version of the paper.
Line 88 “please homogenize the font size.”
Answer: We homogenised the font size for the entire document. .
Line 99 “please change 'PNPG' to 'the north western Portugal (PNPG)'”
Answer: We changed according to the reviewer’s suggestion (line 103).
Line 109 'how they may affect'
Answer: We changed according to the reviewer’s suggestion (line 114).
Line 112 'these PNPG populations'
Answer: We changed to “the PNPG population”, following the reviewer’s suggestion (line 117-118).
Line 114 ‘please add the bracket’
Answer: We changed according to the reviewer’s suggestion (line 120).
Line 172 please, specify 'number of migrants', since this abbreviation does not appear in the quoted article
Answer: We specified the meaning of the expression Nm. The reviewer is correct and – contrarily to the R package used in this analysis – the original paper does not refer to Nm.
Line 307 'Celtic Iberian haplotypes that were also...' ?
Answer: We changed according to the reviewer’s suggestion (line 310).
Figure 4 ‘H13 should be in bold characters’
Answer: We changed according to the reviewer’s suggestion.
Line 410 ‘PNPG’
Answer: We changed according to the reviewer’s suggestion (line 408).
Line 418 ‘I suppose this is only true for microsatellite nuclear markers’
Answer: We changed according to the reviewer’s suggestion (line 417).
Line 493 ‘British’
Answer: We changed the references according to the journal requirements..
Line 506 ‘Hewison’
Answer: We changed the references according to the journal requirements
Line 508 ‘PLoS One’; ‘comma or colon?’
Answer: We changed the references according to the journal requirements
Line 517 ‘add the commas’
Answer: We changed the references according to the journal requirements
Line 510 ‘add the co-authors’
Answer: We changed the references according to the journal requirements
Line 520 ‘resource-defense’
Answer: We changed the references according to the journal requirements
Line 527 ‘A.J.M.’
Answer: We changed the references according to the journal requirements
Line 541 ‘Journal of Zoology’
Answer: We changed the references according to the journal requirements
Line 577 ‘DOI Not Found’
Answer: We changed the references according to the journal requirements We corrected it in the paper, and the correct DOI is: https://dx.doi.org/10.2305/IUCN.UK.2016-1.RLTS.T42395A22161386.en
Line 578 ‘in italics’
Answer: We changed the references according to the journal requirements
Line 613 ‘Review’
Answer: We changed the references according to the journal requirements
Line 623 ‘in italics’; ‘remove space’
Answer: We changed the references according to the journal requirements
Line 627 ‘add commas’
Answer: We changed the references according to the journal requirements
Line 640 ‘in italics’
Answer: We changed the references according to the journal requirements
Line 646 ‘please, check the references again, I may have missed other errors’
Answer: The references were checked and corrected in the revised version of the manuscript.
Reviewer 2 Report
Review for animals-1018935
The manuscript presents the analysis or reanalysis of microsatellite and SNP data from roe deer in the Iberian peninsula to demonstrate that in Portugal there is both a Quaternary relict population and a recently introduced population from Central Europe (which used to be a source to stock private hunting grounds during the XXth century).
I am not a geneticist at all but I accepted to review the ms because of my experience with the species. Hence I am only commenting on the interpretation/communication aspects, I am not validating the methods.
- The title is convoluted for no reason. No need to formulate it as a question. For example: “Portugal roe deer are a diverse mix of a newly identified Quaternary relict population, contributions from other Iberian refugia, and a recently introduced population from Central Europe”. The key words admixture, allochronous can go in the keyword section.
- The introduction could be made more impactful by starting with the specificities of the European continent vs. other landmarks, which have had a structuring effect on its biodiversity (dominance of a rather small ungulate, multiple refugia in the peninsulas separated by E-W mountain ranges etc).
- The conservation implication of the findings should be mentioned in the abstract: people should not translocate roe deer into the relict population (that is not only the national park, but all the region surrounding it). Whenever possible, hunting efforts should be directed towards the peripheral areas instead of the central areas to prevent immigration of allochtonous deer.
- I believe the most up to date statistical methods now provide a cluster assignment probability based on both mitochondrial and nuclear loci. It’s a bit weird to provide two sets of separate results in my opinion.
- The authors should provide a zoomed map of the relief in the study region to showcase the mountain ranges that isolated the refugia.
- The authors should provide some estimate of the time since divergence between the newly identified relict and other populations.
Author Response
Reviewer 2
The manuscript presents the analysis or reanalysis of microsatellite and SNP data from roe deer in the Iberian Peninsula to demonstrate that in Portugal there is both a Quaternary relict population and a recently introduced population from Central Europe (which used to be a source to stock private hunting grounds during the XXth century).
Answer: We thank the reviewer for his insightful comments and suggestions, that helped us to improve the paper. Because we believe that there might have been some misinterpretation on the type of genetic data we used and that it might have influenced some comments downstream, we would only like to call attention to the fact that our data set consists of genotypes based on microsatellite data and sequence data pertaining to the d-loop region of the mitochondrial chromosome. We did not generate SNP (single nucleotide polymorphism data) nor the length of the generated sequence allow us to treat point mutations as SNPs, for polymorphisms in the sequence are much likely linked and are not independently segregated.
I am not a geneticist at all but I accepted to review the ms because of my experience with the species. Hence I am only commenting on the interpretation/communication aspects, I am not validating the methods.
- The title is convoluted for no reason. No need to formulate it as a question. For example: “Portugal roe deer are a diverse mix of a newly identified Quaternary relict population, contributions from other Iberian refugia, and a recently introduced population from Central Europe”. The key words admixture, allochronous can go in the keyword section.
Answer: Our aim, with this title, was to allow the title to provide a first glimpse on some of the hypothesis being tested and also the main results. The question at the beginning would raise the questions: (i) ancient: if PGNP population was actually a relict population; (ii) admixed: to what extent, roe deer populations in the region were admixed? (iii) allochthonous: if roe deer in central and southern Portugal was actually mainly descent from a non-native gene pool introduced in the 90s? The second part of the title would provide the main answer: that several populations, with distinct origins, exist in the region.
However, we do understand the reviewer’s criticism and admit that a shorter title could work better. We feel that the alternative kindly presented by the reviewer might not solve the problem for it is even eight words longer than the original title. On the other hand, it might erroneously suggest that the relict population of PGNP was first identified by this study but it was actually first suggested by Randi et al (2004).
Therefore, if the editor and the reviewer do prefer that we change the title of the paper, we will gladly accept the suggestion but present a shorter, alternative title, based on the original one:
“The multiple origins of roe deer populations in Western Iberia and their relevance for conservation”.
- The introduction could be made more impactful by starting with the specificities of the European continent vs. other landmarks, which have had a structuring effect on its biodiversity (dominance of a rather small ungulate, multiple refugia in the peninsulas separated by E-W mountain ranges etc).
Answer: We tried to keep the introduction within a reasonable scope and length, given the aims of this paper. While extensive literature on the subject has been published in the last years, we agree with the reviewer that a wider scope might be presented at the beginning of the introduction, in order to place this contribution within the context of studies focusing on the role of southern European peninsulas as refugia and centers of biological diversification. Thus, we accepted the suggestion and incorporated the following paragraph at the beginning of our introduction, starting from a larger scope and narrowing it until focusing on roe deer intraspecific diversity. We hope that the introduction is now more impactful and adequate. Nevertheless, we are available for additional modifications, if necessary. The introduction now starts with the following paragraph:
“The Mediterranean Basin is one of the very few World’s biodiversity hotspots located outside the tropics (Myers et al., 2000). While the convergence of two biogeographic realms (Holt et al., 2013) contributes to this region’s singularity, the interaction among Earth’s climatic cycles and Europe’s plays a crucial role. At higher latitudes, Europe is a continuous landmass, but its southern ranges are composed of three – Iberian, Apennine and Balkan – peninsulas, isolated, at some extent by mountain ranges along an E-W axis. It is nowadays well accepted that these peninsulas played a role as refugia and diversification centers, both at the inter and intraspecific level (Weiss & Ferrand, 2007), namely for large mammal species (Randi 2007), such as carnivores and ungulates.”
- The conservation implication of the findings should be mentioned in the abstract: people should not translocate roe deer into the relict population (that is not only the national park, but all the region surrounding it). Whenever possible, hunting efforts should be directed towards the peripheral areas instead of the central areas to prevent immigration of allochtonous deer.
Answer: We thank the reviewer for this suggestion. In fact, we did explicitly refer to these conservation issues in a first version of our abstract. However, due to space limitations (journal guidelines refer a maximum of 250 words in the abstract), we had to remove it from the abstract for we found no other place to cut from. We now include again references to conservation implications in the abstract. However, the abstract is now 270 words long. We checked on recently published abstracts in “Animals”, and found a few with more than that (https://doi.org/10.3390/ani10010169; https://doi.org/10.3390/ani10010165). In a particular case, more than 300 words (https://doi.org/10.3390/ani10010172). If we are allowed by the editors to present a 270-word abstract, we will be glad to accept the reviewer’s suggestion. We edited the abstract accordingly and it now reads as follows:
“The roe deer (Capreolus capreolus) is native and widespread in Europe and its phylogeography has been clarified in the last decades. Southern peninsulas are considered as reservoirs of genetic diversity and the source for the recolonization of Europe after the last glacial maximum. Even though roe deer populations have been genetically characterized, there is a major knowledge gap about the populations at the western edge of its distribution. To fill this caveat, and based on mitochondrial and nuclear DNA data, we aim to: i) characterize the genetic diversity and structure of roe deer in western Iberia; ii) clarify the origins and phylogeographical affinities of these populations, namely the relict population from Peneda Gerês National Park (PGNP, Portugal) and the likely allochthonous populations from central and south (CS) Portugal; iii) discuss the implications of our findings for the management and conservation of the roe deer. Three major genetic clusters were inferred based on nuclear genotypes and were structured in a similar way as the three major mtDNA clades present in Iberia. Patterns inferred with nuclear markers confirmed PGNP as a relict population. Roe deers from CS Portugal share haplotypes with Central Europe rather than with other western Iberian populations, confirming its mainly allochthonous origin. Our results highlight western Iberia as a diversity hotspot for roe deer. We highlight the role of intraspecific genetic diversity as a source of resilience against ongoing global changes; the need for transboundary management and the importance of genetic data to inform management and conservation. When considered, repopulation or translocation measures should follow the IUCN Law of Reintroductions and meticulously conducted in order to preserve the genetic heritage of the species.”
We only refrain from proposing measures regarding hunting of roe deer, because this hunting is very residual in Portugal, and is not even allowed in PGNP.
- I believe the most up to date statistical methods now provide a cluster assignment probability based on both mitochondrial and nuclear loci. It’s a bit weird to provide two sets of separate results in my opinion.
Answer: The reviewer is correct in stating that there are current methods that allow for the simultaneously use of mitochondrial and nuclear loci for cluster assignment. However, we fear that the suggestion is partly influenced by a misunderstanding of the nature of our original data. As mentioned above, we do not have data on nuclear microsatellite and mitochondrial SNPs. We have nuclear microsatellite and mtDNA sequence data. Most of the methods the reviewer is referring to use simultaneously microsatellites and SNPs. Only very few – to our knowledge – use genotype and haplotype (sequence) data simultaneously. One example is DYIABC, but is not focused on cluster assignment. One of the reasons is because the best approaches for inferring structure from microsatellite data are based on Bayesian algorithms that optimize parameters and criteria (such as Hardy-Weinberg equilibrium) that are not paralleled in non-recombinant haplotype (sequence) data. Clustering algorithms based on mtDNA haplotypes optimizes different criteria such as allele (haplotype) frequency in each population. Additionally, merging the two datasets would prevent us from inferring several parameters that are ecologically meaningful in the context of one or other marker (such as evidence for bottlenecks, heterozygosity or inbreeding coefficients in one hand, and nucleotide diversity or average number of mutations on the other).
In the end, analysing both datasets separately, we have access to two different stories on the populations: a story about the females, female philopatry and dispersion, that is told by mtDNA (non-recombinant, matrilineal) markers; another story about the whole population, told by nuclear (recombinant and highly polymorphic) microsatellite markers, that more accurately estimates gene flow, admixture and population structure and dynamics. We call attention to the fact that, even not knowing the sex of individuals, male mammals are (with very few exceptions) dead ends in what concerns mtDNA chromosome inheritance. Thus, mtDNA patterns always reflect female dynamics and movements in a population or populations. Therefore, even if we could analyse merge the two datasets and analyse it together, we would likely prefer not to do it, in order to obtain evidence from two different kinds of markers that reflect different stories.
- The authors should provide a zoomed map of the relief in the study region to showcase the mountain ranges that isolated the refugia.
Answer: We thank the reviewer for this comment and proceeded accordingly. We have produced a new map depicting the major mountain chains and rivers in Iberia – that might have had a role in the differentiation of roe deer populations. We considered the chance of including this information directly in figure 1 but with the need to incorporate information on genetic diversity (pie charts), we feared that the most relevant geographic features would be omitted by the reporting of results. Therefore, we produced this new figure, incorporating also the information on the location of sampling points within each sampling region. We now think this figure is more informative and replaced current figure S1 (supporting material) by the new one, since it also includes information on sample size. If the reviewer and editor find it useful, we are open to include this figure as figure S1 in the paper, and shift figure numbering, for all other figures.
The new figure is as follows.
- The authors should provide some estimate of the time since divergence between the newly identified relict and other populations.
Answer: Estimation of time since divergence has become increasingly easier in the context of species (phylogeography) and populations (population genetics) divergence, the and a large diversity of methods has emerged, in particular since the coalescent theory has been formalized and implemented in software for genetic analysis. There is however, a major limitation of these methods, when it comes to the estimation of divergence of populations such as the ones in our study. Approaches using coalescent theory often assume reciprocal monophyly among populations. That is, individuals from one population are assumed to be evolutionary closer to each other than to any other individual in a different population. In cases were migration and admixture are frequent, these methods tend to perform badly (an excellent discussion on these methods is offered by Arbogast et al, 2002). Our case is even worst for our populations are not reciprocally monophyletic to each other. In fact, all populations include individuals from different mitochondrial genetic clades. Also, most (microsatellite) alleles are shared by at least two populations. To our knowledge, there is no method that could provide minimally accurate divergence time between the populations under study. Divergence times could, however, be estimated for the lineages that are dominant in each populations: Central Clade (dominant in Central and South Portugal, and in Trás-os-Montes and Galicia, though very distinct haplotypes); Celtic-Iberian Clade (dominant in Astúrias and Valsemana) and Clade West (dominant in PGNP, and also in South Spain, as reported by Royo et al (2007) and Randi et al (2004). The divergence times among these clades or lineages might be a good surrogate for estimating the time of divergence of the lineages that might be at the root of these populations. These divergence times were already estimated independently by Royo et al (2007) and by Lorenzini et al (2014). While the estimation of divergence times departs from the main focus of this paper, the reviewer is correct in highlighting their relevance for understanding the importance of these relict populations. Thus, we have incorporated in our discussion references on the divergence time among the major lineages present in NW Iberia, in order to highlight their singularity. We included the following lines in the discussion:
“Northwestern Iberian populations are dominated by haplotypes from three major (Central, West and Celtic-Iberian) clades. Estimated divergence times among the three clades largely precedes the last glacial maximum, ranging from 98-35kya or 91-45kya – betwen Central and West or Celtic-Iberian clades, respectively – to 221-111kya – between Celtic Iberian and West clade (Royo et al., 2007; Lorenzini et al., 2014). Such an old diversification suggests that roe deer might have arrived and diversified within Iberia more than once.”
We hope that our reply and the inclusion of this information in the paper will be considered sufficient by the reviewer. If you consider that estimation of divergence times is of utmost importance for this paper, we will try to incorporate these estimates – despite the limitations addressed above – but you will have to grant us a much more extended deadline for such analysis are necessarily complex and time consuming.
References:
Arbogast, B.S., Edwards, S.V., Wakeley, J., Beerli, P., Slowinski, J.B. (2002). Estimating divergence times from molecular data on phylogenetic and population genetic timescales. Annual Review on Ecology, Evolution and Systematics, 33:707-740.
Lorenzini, R., Garofalo, L., Qin, X., VOloshina, I., Lovari, S. (2014). Global phylogeography of the genus Capreolus (Artiodactyla: Cervidae), a Palaearctic meso-mammal. Zoological Journal of the Linnean Society, 170:209-221.
Randi, E., Alves, P.C., Carranza, J., Milosevic-Zlatanovic, S., Sfougaris, A., Mucci, N. (2004). Phylogeography of roe deer (Capreolus capreolus) populations: the effects of historical genetic subdivisions and recent nonequilibrium dynamics. Molecular Ecology, 13, 3071–3083.
Royo, L.J., Pajares, G., Álvarez, I., Fernández, I., Goyache, F. (2007). Genetic variability and differentiation in Spanish roe deer (Capreolus capreolus): A phylogeographic reassessment within the European framework. Molecular Phylogenetics and Evolution, 42, 47–61.